# A New Classification System to Predict Functional Outcome after Laryngectomy and Laryngopharyngectomy

**DOI:** 10.3390/cancers13061474

**Published:** 2021-03-23

**Authors:** Stefan Grasl, Elisabeth Schmid, Gregor Heiduschka, Markus Brunner, Blažen Marijić, Matthaeus Ch. Grasl, Muhammad Faisal, Boban M. Erovic, Stefan Janik

**Affiliations:** 1Department of Otorhinolaryngology, Head and Neck Surgery, Medical University of Vienna, 1090 Vienna, Austria; stefan.grasl@meduniwien.ac.at (S.G.); elisabeth.a.schmid@meduniwien.ac.at (E.S.); gregor.heiduschka@meduniwien.ac.at (G.H.); markus.brunner@meduniwien.ac.at (M.B.); Matthaeus.grasl@meduniwien.ac.at (M.C.G.); 2Institute of Head and Neck Diseases, Evangelical Hospital, 1180 Vienna, Austria; blazen.marijic@uniri.hr (B.M.); b.erovic@ekhwien.at (B.M.E.); 3Department of Otorhinolaryngology, Head and Neck Surgery, Clinical Hospital Center Rijeka, 51000 Rijeka, Croatia; 4Shaukat Khanum Memorial Cancer Hospital, Lahore 54000, Pakistan; muhammadfaisal@skm.org.pk

**Keywords:** laryngectomy, laryngopharyngectomy, salvage surgery, reconstruction, functional outcome, classification system

## Abstract

**Simple Summary:**

Evaluation of the long-term functional outcome after primary or salvage laryngopharyngectomy. Long term functional outcome mainly depends on extent of pharyngectomy and salvage situation, which is reflected by our new classification system.

**Abstract:**

(1) Objective: To evaluate long-term functional outcome in patients who underwent primary or salvage total laryngectomy (TL), TL with partial (TLPP), or total pharyngectomy (TLTP), and to establish a new scoring system to predict complication rate and long-term functional outcome; (2) Material and Methods: Between 1993 and 2019, 258 patients underwent TL (*n* = 85), TLPP (*n* = 101), or TLTP (*n* = 72). Based on the extent of tumor resection, all patients were stratified to (i) localization I: TL; II: TLPP; III: TLTP and (ii) surgical treatment (A: primary resection; B: salvage surgery). Type and rate of complication and functional outcome, including oral nutrition, G-tube dependence, pharyngeal stenosis, and voice rehabilitation were evaluated in 163 patients with a follow-up ≥ 12 months and absence of recurrent disease; (3) Results: We found 61 IA, 24 IB, 63 IIA, 38 IIB, 37 IIIA, and 35 IIIA patients. Complications and subsequently revision surgeries occurred most frequently in IIIB cases but rarely in IA patients (57.1% vs. 18%; *p* = 0.001 and 51.4% vs. 14.8%; *p* = 0.002), respectively. Pharyngocutaneous fistula (PCF) was the most common complication (33%), although it did not significantly differ among cohorts (*p* = 0.345). Pharyngeal stenosis was found in 27% of cases, with the highest incidence in IIIA (45.5%) and IIIB (72.7%) patients (*p* < 0.001). Most (91.1%) IA patients achieved complete oral nutrition compared to only 41.7% in class IIIB patients (*p* < 0.001). Absence of PCF (odds ratio (OR) 3.29; *p* = 0.003), presence of complications (OR 3.47; *p* = 0.004), and no need for pharyngeal reconstruction (OR 4.44; *p* = 0.042) represented independent favorable factors for oral nutrition. Verbal communication was achieved in 69.3% of patients and was accomplished by the insertion of voice prosthesis in 37.4%. Acquisition of esophageal speech was reached in 31.9% of cases. Based on these data, we stratified patients regarding the extent of surgery and previous treatment into subgroups reflecting risk profiles and expectable functional outcome; (4) Conclusions: The extent of resection accompanied by the need for reconstruction and salvage surgery both carry a higher risk of complications and subsequently worse functional outcome. Both factors are reflected in our classification system that can be helpful to better predict patients’ functional outcome.

## 1. Introduction

Total laryngectomy (TL) followed by radiotherapy (RT) represented the standard treatment for patients with locally advanced laryngeal cancer for almost a century. The release of the landmark veterans’ affairs trial in 1991 reporting of similar oncological outcome in patients with induction chemotherapy followed by RT while preserving the larynx represented the starting point for various types of organ-preservation protocols [1,2,3]. 

The stigma of permanent tracheostomy and the loss of natural voice are powerful drivers for many patients to dismiss primary laryngectomy [4]. However, not all patients are suitable candidates for organ-preserving techniques. Especially, patients with extensive extralaryngeal or transglottic extension and those with poor laryngeal function with significant impairments of airway and/or swallowing will not benefit from organ preservation [3]. This differentiation is mandatory as even in highly selected patient cohorts, salvage laryngectomy is necessary in 25% to 36% of cases due to missing response or locoregional recurrence [1,5]. 

Moreover, attempts to preserve larynx in hypopharyngeal carcinomas by organ-preservation protocols are less encouraging due to more frequent loco-regional recurrences ending in partial (TLPP) or total laryngopharyngectomy (TLTP) that are associated with worse outcome compared to primary laryngopharyngectomy [6].

Although salvage surgery provides adequate oncological outcome with 5-year overall survival rates of 30% to 70%, it is also associated with overall complications in up to 67.5% of cases [7,8]. Among them, pharyngocutaneous fistula (PCF) is the most common (28.9%) followed by dysphagia (18.6%) and pharyngeal stenosis (14.3%) [7,9,10,11], which either require surgical revision or reconstruction with vascularized regional or free flaps and at least represent a significant impairment of patient’s quality of life. The importance of assessment of voice and swallowing as functional parameters in treatment decision-making has been previously highlighted. Despite organ preservation, severe dysphagia accompanied by the risk of aspiration and consequently pneumonia are inherent limitations and major concerns of patients [12]. 

Hence, we strongly believe that functional parameters need stronger consideration in the context of personalized medicine. Therefore, the main purpose of the study was to evaluate long-term functional outcome in patients undergoing primary TL, TLPP, and TLTP compared to those undergoing salvage procedures. 

## 2. Materials and Methods

### 2.1. Study Cohort

We performed a retrospective cohort study of 258 patients who underwent treatment for advanced staged squamous cell carcinoma (SCCs) of the larynx, hypopharynx, or hypopharynx with invasion of the cervical esophagus (*n* = 255) or chondrosarcoma (*n* = 2) and chondroma (*n* = 1) of the larynx. All patients were treated between January 1993 and October 2019 at the Department of Otorhinolaryngology, Head and Neck surgery of the Medical University of Vienna and the Institute of Head and Neck diseases, Evangelical Hospital Vienna. In particular, 92 patients (35.7%) were treated in the 1990s (<2000), 79 (30.6%) were treated between 2000 and 2009, and 87 (33.7%) were treated between 2010 and 2019. 

Clinical data were retrospectively obtained from electronic patient records. Data were collected regarding basic patient details (sex, age, smoking behavior, Charlson Comorbidity Index [13], body mass index (BMI)), surgical details, such as extent of ablative surgery (laryngectomy vs. laryngectomy with partial pharyngectomy vs. laryngectomy with total pharyngectomy), type of pharyngeal reconstruction (primary closure vs. flap reconstruction vs. jejunum reconstruction) and whether or not regional or free flaps were used in a primary or salvage setting. Total pharyngectomies were defined as circumferential pharyngeal resections, while any amount of pharyngeal resection less than circumferential defects, comprising primary closures as well as epithelialized inlay flaps, were defined as partial pharyngectomies. Moreover, tumor characteristics including primary tumor site and tumor staging were evaluated. 

### 2.2. The Vienna Laryngopharyngectomy Classification System

Based on primary and salvage surgery, we stratified our patients into six subgroups that were assessed regarding functional outcome and complications: (a) Type of surgery [I: laryngectomy (TL); II: laryngectomy with partial pharyngectomy (TLPP); III: laryngectomy with total pharyngectomy (TLTP)]; (b) Salvage surgery (A: non-salvage; B: salvage).

### 2.3. Functional Endpoints

Functional outcomes were evaluated in patients with follow-ups longer than 12 months and absence of recurrent disease during the first year of follow-up. Voice rehabilitation and swallowing represented the main functional endpoints.

We differentiated whether patients were able to verbally communicate or not and whether voice restoration was accomplished by (i) voice prosthesis, (ii) esophageal speech, or (iii) usage of an electro larynx. Those patients who were unable to verbally communicate or just whispered were allocated to the electro larynx subgroup. We further differentiated whether a voice prosthesis was inserted primarily during laryngectomy or secondarily months after the surgical intervention.

With regard to swallowing outcome, patients were classified as being completely gastrostomy tube dependent (no intake per mouth), having a combination of feeding tube and oral intake (partial oral nutrition), or being able to have unrestricted oral intake alone (total oral nutrition) at last time of follow-up. 

Moreover, the development of pharyngeal stenosis and subsequent need for esophageal dilation defined as any attempted esophageal dilatation in the operating room represented another functional endpoint. Modified barium swallow (MBS) studies were postoperatively performed during the 10th–15th postoperative day (POD) either to confirm patency of the pharynx and absence of leaks and subsequently fistulas (i) or to identify the location and extent of pharyngeal stenosis (ii). Patients who were diagnosed with a pharyngeal stenosis and loco-regional recurrences were excluded from functional analyses. 

### 2.4. Complications

Medical records were further reviewed regarding revision surgeries and complications associated with laryngectomy or laryngopharyngectomy, such as pharyngocutaneous fistula (PCF), the development of pharyngoesophageal stenosis, loco-regional or donor site wound dehiscence, swallowing deficits, flap failure, or hemorrhage requiring surgical revision. PCF was defined either as a (i) clinically manifest salivary leakage through dehiscent skin or mucosa or (ii) small leakage, not clinically obvious, just found on the MBS study. 

Systemic infections (e.g., pneumonia, sepsis, etc.), swelling (e.g., hematoma, chyle, or seroma formations) and severe general medical disorders (e.g., stroke, cardiopulmonary reanimation, or myocardial infarction) were defined as complications that occurred between surgery and discharge. Concurrent and/or consecutive complications were classified as multiple complications. 

### 2.5. Statistical Methods

Statistical analyses were performed using SPSS version 27.0 software (IBM SPSS Inc., Armonk, NY, USA). Unless otherwise specified, data are reported as mean ± standard deviation (SD). Descriptive statistics were used for analysis of demographic and clinical data. The Chi-square test was used to investigate the association between nominal variables. An unpaired Student’s t-test was used to compare means of two independent groups with normal (Gaussian) distributions. Kaplan–Meier analyses and Log-rank test were assessed for univariate outcome analysis. Uni- and multivariate binary logistic regression analyses were used to evaluate the prognostic impact of different clinical variables on functional endpoints including pharyngeal stenosis, complete oral nutrition, G-tube dependence, and occurrence of PCF. Odds ratios (ORs) and corresponding 95% confidence intervals (CIs) are indicated. All tests were performed two-sided, and *p*-values below 0.05 were considered statistically significant.

## 3. Results

### 3.1. Patient Population

In total, 258 patients were analyzed comprising 230 (89.1%) males and 28 (10.9%) females with a mean age of 59.2 ± 9.2 years. Tumors were more commonly located in the larynx (53.5%) than in the hypopharynx (46.5%). Ten (3.9%) T1, 33 (12.8%) T2, 88 (34.1%) T3, and 117 (45.3%) T4 tumors were noted with positive neck nodes in 123 (47.7%) patients (Table 1). Three (1.2%) patients underwent TL due to advanced chondroma/chondrosarcoma, and seven (2.7%) patients required laryngectomy because of a dysfunctional larynx after primary RT. Subsequently, those 10 patients were not assessed in TNM staging. In total, 97 (37.6%) patients underwent salvage and 161 (62.4%) underwent primary surgery. Altogether, 143 (55.4%) patients underwent postoperative radiotherapy (PORT), including eight patients additionally receiving adjuvant chemo- or immunotherapy. Further socio-demographic data with regard to primary or salvage surgery are shown in Table 1.

### 3.2. Surgical Ablation

TLPP was performed in 101 (39.14%) cases followed by TL only and TLTP in 85 (32.9%) and 72 (27.9%) cases, respectively. Neck-Dissection (ND) was done in 200 (77.5%) patients and significantly more often during primary surgery (97.5% vs. 44.3%; *p* < 0.001). TL was significantly more commonly performed in primary surgeries (37.9% vs. 24.7%), while TLTP was more likely done in salvage settings (36.1% vs. 23%; *p* = 0.032). 

### 3.3. Reconstruction

Pharyngeal closure after total laryngectomy with partial pharyngeal defects (TLPP) was achieved either with (i) a primary closure [83 (82.2%) patients], (ii) a free flap [4 (4%) patients], a jejunum [2 (2%) patients], or (iii) a pedicled pectoralis muscle flap (12 (11.9%) patients). Conversely, total (circumferential) pharyngeal defects were most commonly reconstructed with a jejunum in 55 (76.4%), followed by free flap in 10 (13.9%) and pectoralis major myocutaneous flap (PMMF) in seven (9.7%) cases. In particular, the anterolateral thigh (ALT) flap was used in 8.3% (*n* = 6), the radial forearm free flap (RFFF) was used in 4.2% (*n* = 3), and the serratus anterior free flap (SAFF) was used in 1.4% (*n* = 1) of cases for total pharynx reconstruction. Moreover, the SAFF, ALT, and RFFF were used for partial pharynx reconstruction in 2% (*n* = 2), 1% (*n* = 1), and 1% (*n* = 1), respectively. The PMMF represented the workhorse flap in our cohort, which was applied in 35.3% of cases (*n* = 91) either combined with jejunal free transfer (*n* = 21) or RFFF (*n* = 3), alone (*n* = 67), as a pharyngeal patch (*n* = 21), or for wound closure (*n* = 70). Salivary tube was used in 25 patients (14.5%) after laryngopharyngectomy and particularly in salvage surgeries (*p* < 0.001). Applied flaps and whether they were used either for primary or revision surgery or pharynx reconstruction and/or wound closure are depicted in Figure 1. 

### 3.4. Complications According to the Proposed Classification System

Complications occurred in 32.2% of cases (*n* = 85) and were significantly more common in salvage procedures (43.3% vs. 24.8%; *p* = 0.002). Consequently, revision surgeries were also more frequently done in salvage cases compared to patients treated upfront (39.2% vs. 19.3%; *p* < 0.001) with a median time between initial and revision surgery of 40.5 days. Moreover, complications were also significantly less in patients suitable for PORT representing non-salvage procedures with uneventful postoperative courses (40.2% vs. 59.8%; *p* = 0.001). It is noteworthy that the complication rate did not significantly change among the last decades (*p* = 0.167). However, reasons for revision surgery were PCF (*n* = 30), hemorrhage (*n* = 16), wound-healing deficits (*n* = 10), free flap failure (*n* = 6), and pharyngeal stenosis (*n* = 5). With regard to our supposed classification system, complications occurred most commonly in TLTP patients (class III) compared to TLPP (class II) and TL (class I), respectively (43.1% vs. 34.7% vs. 18.8%; *p* = 0.004). Moreover, non-salvage (IA) and salvage TL (IB) carried the lowest risk of complications (18% and 20.8%) followed by non-salvage TLPP (IIA) and non-salvage TLTP (IIIA) with complications in 28.6% and 29.7% of cases. Salvage TLPP (IIB) and TLTP (IIIB) had a two-times (44.7%) and almost three-times (57.1%) higher risk of complications compared to TL (*p* = 0.001; Table 2), respectively (Figure 2).

Two patients deceased after salvage laryngopharyngectomy due to cardiovascular failure and tumor progression after an incomplete tumor resection resulting in an overall mortality rate of 0.78%.

### 3.5. Pharyngocutaneous Fistula

PCF was diagnosed in 85 out of 258 (33%) patients. Among them, 67 PCFs occurred within the 30th POD, while 18 cases occurred thereafter (78.8% vs. 21.2%). The median time between surgery and occurrence of PCF was 15.0 days (25th–75th percentile 10–27 days). Interestingly, except for female gender (*p* = 0.046; OR= 2.24), none of the tested clinical variables, including salvage versus primary surgery or primary closure versus pharyngeal closure significantly influenced the overall risk of developing PCFs (Table 2 and Table 3). Figure 3 illustrates the potential relationships and nexus between PCF formation, other short-term complications, and flap usage in the entire collective (A) and for salvage procedures separately (B). Altogether, it becomes obvious that the necessity of pharyngeal reconstruction increases morbidity and consequently causes higher rates of complications and PCF formation. 

### 3.6. Functional Outcome

After exclusion of all patients with incomplete tumor resection (*n* = 7), those with a follow-up period of less than one year (*n* = 19), and those who experienced recurrent disease or deceased within one year after laryngectomy or laryngopharyngectomy (*n* = 75), 163 patients were finally available for functional analyses. 

The mean follow-up time for analysis of functional endpoints was 69.7 months with a range of 12 to 300 months. The mean patient age of the cohort was 59.1 ± 8.8 y with a female to male ratio of 11% to 89%, which was identical compared to the whole cohort. However, the preoperative BMI was significantly lower in females (20.6 ± 4.4 kg/m^2^ vs. 24.3 ± 4.2 kg/m^2^; *p* = 0.002), while the rate of laryngopharyngectomies was significantly higher (50% vs. 25.8%; *p* = 0.008) compared to males. As a consequence, pharyngeal reconstruction was relatively more often performed in female patients (*p* = 0.019). 

### 3.7. Swallowing

At the end of follow-up, 71.2% (*n* = 116) of patients were capable of unrestricted oral nutrition, and only 4.9% (*n* = 8) were dependent on gastrostomy tube feeding. The remaining 23.9% (*n* = 39) of cases were able to have at least partial oral nutrition. 

### 3.8. Temporary Gastrostomy Tube Dependence

A subset of 23 out of 142 patients (16.2%) was temporally dependent on G-tube feeding during therapy and 21 patients needed a G-tube due to recurrence within the first year after surgery. The presence of stenosis (*n* = 14), PCF (*n* = 5), stenosis, and PCF (*n* = 2) were linked to swallowing impairment that finally required G-tube insertion. After MBS study, no reason could be found to explain dysphagia in two patients. The cases with carcinomas primarily originating from the larynx (OR 0.27; *p* = 0.006), no necessity for free flap reconstruction (OR 0.21; *p* = 0.001), primary pharyngeal closure (OR 0.21; *p* = 0.001), and absence of PCF (OR 0.20; *p* = 0.001) carried a significantly lower risk of temporary G-tube dependence, while females had a significantly higher probability. Altogether, only absence of PCF represented in multivariate analysis an independent prognosticator associated with a four-times lower risk (OR 0.21; *p* = 0.004) for G-tube dependence (Table 4). 

### 3.9. Pharyngeal Stenosis

Pharyngeal stenosis causing functional impairment was noticed in 40 out of 148 patients (27.0%) with a mean time between surgery and occurrence of stenosis of 18.8 ± 25.1 months. Moreover, 15 patients developed stenosis due to recurrence within the first year after surgery.In particular, 24 (60%), 32 (80%), and 36 (90%) stenosis cases occurred during the first, second, and fifth year after surgery, respectively. Dilatation was performed in 37 out of 40 (92.5%) cases. Among them, 14 (37.8%) and 16 (43.2%) patients became capable of complete or partial oral nutrition, respectively. 

Stenosis was detected more commonly in hypopharyngeal (42.6%) than in laryngeal carcinomas (16.1%; *p* = 0.001). Stenosis occurred in 55.4% of patients undergoing TLTP, which was significantly higher compared to 29.3% and 8.8% in patients undergoing TLPP and TL (*p* < 0.001), respectively. The importance of tumor site and subsequently extent of resection on occurrence of stenosis is reflected by binary logistic regression analysis showing, among others, that laryngeal carcinomas (OR 0.26; *p* = 0.001) and cases with primary pharyngeal closure (OR 0.18; *p* < 0.001) carried the lowest risk of stenosis. Moreover, stenosis was found more often in patients with lower preoperative BMI (OR 2.97; *p* = 0.014) and females (OR 4.81; *p* = 0.003). As shown in Figure 4, the occurrence of stenosis was not statistically significant different between primary and salvage surgery (*p* = 0.975) and cases with and without pharyngeal reconstruction (*p* = 0.525). 

### 3.10. Oral Nutrition

The probability for complete oral nutrition was significantly higher in patients without the need for pharyngeal reconstruction (OR 4.72; *p* < 0.001), in non-salvage surgeries (OR 2.42; *p* = 0.015), uneventful perioperative courses (OR 3.56; *p* = 0.001), and absence of PCF (OR 3.13; *p* = 0.002) during follow-up. Accordingly, complete oral nutrition was noticed in 76.3% of laryngeal compared to 63.6% in hypopharyngeal carcinomas (*p* = 0.015), and in 83.9% of cases after TL compared to 71.4% and 50% in patients after TLPP and TLTP (*p* = 0.001), respectively. Absence of PCF (OR 3.26; *p* = 0.004), uneventful postoperative courses without complications (OR 3.46; *p* = 0.004), and no necessity for pharyngeal reconstruction (OR 4.28; *p* = 0.023) represented favorable prognosticators for complete oral nutrition at multivariate analyses (Table 4).

### 3.11. Voice Rehabilitation

Ability for verbal communication was achieved in 69.3% of patients and was accomplished by insertion of voice prosthesis and acquisition of esophageal speech in 37.4% (*n* = 61) and 31.9% (*n* = 52) of patients, respectively. The use of electro larynx, whisper, or absent voice restoration was noticed in the remaining 30.7% (*n* = 50) of cases. Secondary insertion of voice prosthesis (75.4%) was three times more common than insertion within the primary procedure (24.6%). The mean time between primary surgery and secondary insertion of voice prosthesis was 10.6 ± 7.5 months. Primary insertion of voice prosthesis (*n* = 16) was particularly performed in patients undergoing TL (*p* < 0.001), primary surgery (*p* = 0.010), and no need for pharynx reconstruction (*p* = 0.006). However, none of the variables outlined above represented an either positive or negative predictor for voice restoration by using voice prosthesis at multivariate binary logistic regression (data not shown). Interestingly, a voice prothesis was inserted in 15.1% of cases in the 1990s (<2000), which was significantly less compared to 63.5% between 2000 and 2009 and 41.4% between 2010 and 2019 (*p* < 0.001).

### 3.12. Functional Outcome According to the Classification System

Similar to complications, we also assessed functional outcomes with regard to our proposed classification system. The highest rate of functional stenosis was found in class IIIA and IIIB cases undergoing TLTP (53.8% and 75%), while non-salvage TL (class IA) patients showed the lowest incidence of stenosis (8.9%; *p* < 0.001). We found an almost identical association for functional G-tube dependence with the highest dependence rates of 29.4% and 45.5% for class IIIA and IIIB patients (*p* = 0.003). Consequently, the ability for oral nutrition was significantly correlated with our classification system (*p* < 0.001) (Figure 5).

## 4. Discussion

Sufficient oral nutrition and swallowing, uneventful postoperative course and short inpatient stay, voice rehabilitation, and ability for verbal communication are pivotal for patients´ quality of life after removal of their larynx [12]. Consequently, major efforts and progress had been made to improve radiation as well as surgical reconstruction techniques to provide not only best oncological but also functional results [14]. Respecting this fact, we have analyzed the functional outcome of 258 patients who underwent laryngectomy or laryngopharyngectomy and set up a new scoring system based on the complication rate and long-term functional outcome. Our classification system reflects the link between salvage or non-salvage surgery, extent of ablative surgery, and the increasing incidence of complications. In particular, there is a significant gradual decrease of acceptable functional outcome with an extension of ablative resection showing the worst functional outcome particularly in patients with PCF. Class I procedures showed the best swallowing outcome and oral nutrition followed by classes II and III, respectively. For each class, salvage operations showed worse outcome compared to non-salvage surgeries. PCF, occurrence of complications, and particularly the need for pharyngeal reconstruction were strong predictors for persistent parenteral nutrition. Overall, sufficient oral nutrition was accomplished in more than two-thirds of patients, while permanent G-tube dependence was not commonly required. Symptomatically and function-related stenosis causing dysphagia are frequent sequelae after laryngectomy found in approximately one-third of our patients, which is in accordance to previous reported data [7,15,16,17]. In particular, stenosis was noticed in more than half of the patients after TLTP, underlining the importance of tumor site and extent of resection. This is further underlined by our classification system, indicating the highest incidence of stenosis in class III cases followed by class II and class I, respectively.

Our data showed that nearly half of all stenosis (40%) occurred after the first year of surgery. Thus, we hypothesize that wound healing, resulting in centric scar formation, is a dynamic, longer lasting process causing functional impairment and stenosis even years after initial treatment.

Rehabilitation of speech is a major goal after laryngectomy [18]. Indeed, rates for voice rehabilitation range from approximately 80% in patients with laryngectomies to less than half in patients undergoing pharyngectomy with a need for free flap reconstruction [14,19,20]. In our cohort, voice rehabilitation was achieved in more than two-thirds with valved voice prosthesis insertion and sufficiently acquired esophageal speech. Interestingly, in our cohort, sufficient voice rehabilitation was independent of (i) extent of resection, (ii) need for free flap reconstruction, (iii) pretreatment, or (iv) primary tumor location.

Although, several studies already reported on functional endpoints after primary or salvage laryngectomy, laryngopharyngectomy, and on the usage of different flaps, we believe that our study has significant strengths providing new data that adds to the current literature [18,20,21]. First, we have shown long-term functional outcomes of patients with a mean follow-up time of 5 years. Second, our cohort was comprised of total laryngectomies, partial and total laryngopharyngectomies of about one-third each, which underlines the homogeneity of our data. Third, we have shown functional endpoints of patients with primary pharyngeal closures, usage of different fasciocutaneous, and jejunal free transfers, and therefore, our data reflect the whole surgical and reconstructive armamentarium of larynx surgery and reconstruction. However, we see three limiting factors of our study. First, the retrospective character of our study bears an inherent risk of information bias. Second, there is a disproportionate portion of jejunal free transfers for reconstructions that might have influenced functional outcome. Third, the absence of validated scales for oral nutrition, such as the FOIS (Functional Outcome for Oral Intake Scale), represents another drawback of our work [11].

The development of our classification system represents probably the most innovative finding of our study, highlighting two main issues. Salvage procedures do not only carry a significant worse functional outcome compared to primary interventions. However, the risk of complications and subsequently the risk of poor functional outcome increases with the extent of pharyngeal resection and culminating in case of pharyngeal reconstruction. Using our classification system enables the easy illustration of complex information regarding risk factors and functional outcome.

## 5. Conclusions

We could demonstrate that long-term functional outcome is acceptable after laryngectomy and laryngopharyngectomy in primary as well as in salvage settings. The vast majority of patients are capable of total oral nutrition, and permanent G-tube dependence is rare. However, laryngopharyngectomies requiring free flap reconstruction are associated with higher morbidity, complications, and late PCFs that significantly affect the capability of oral nutrition. With the help of a newly created classification system, risks of perioperative complications and estimated functional outcome could be illustrated more easily. Future studies are warranted to validate our new proposed classification system in larger cohorts and prospective settings.

## Figures and Tables

**Figure 1 cancers-13-01474-f001:**
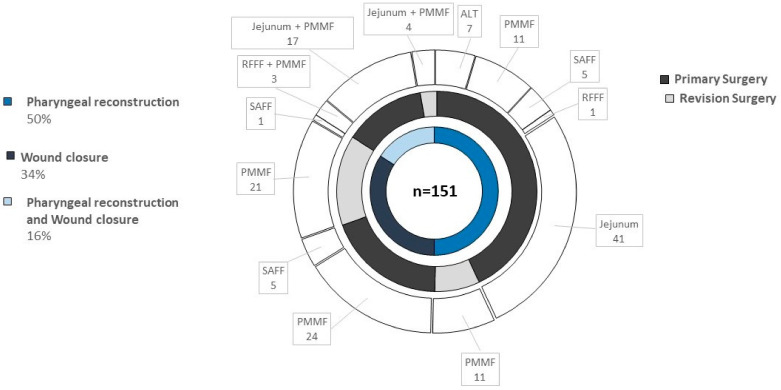
Type of flaps using for reconstruction after laryngopharyngectomy. Graphic representation as sunburst diagram with three ring diagrams. The inner ring displays the type of reconstruction, the middle ring illustrates the time point of reconstruction, and the outer ring shows the type of flaps used for reconstruction.

**Figure 2 cancers-13-01474-f002:**
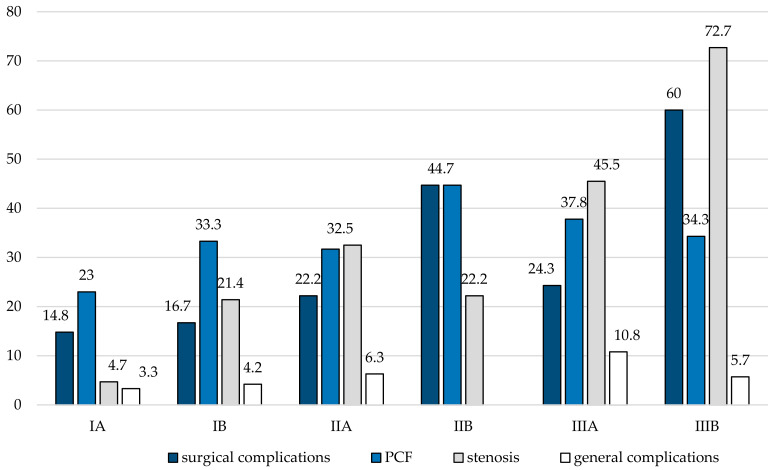
Complications according to the Vienna Laryngopharyngectomy Classification system. Charts showing surgical complications (*p* = 0.001), PCF formation (*p* = 0.345), formation of stenosis (*p* < 0.001), and general complication (*p* = 0.001) depending on the extent of ablative surgery (I: laryngectomy; II: laryngectomy with partial pharyngectomy; III: laryngectomy with total pharyngectomy) and whether each surgery was performed in salvage situation (A: non-salvage; B: salvage). Values are given in percentage.

**Figure 3 cancers-13-01474-f003:**
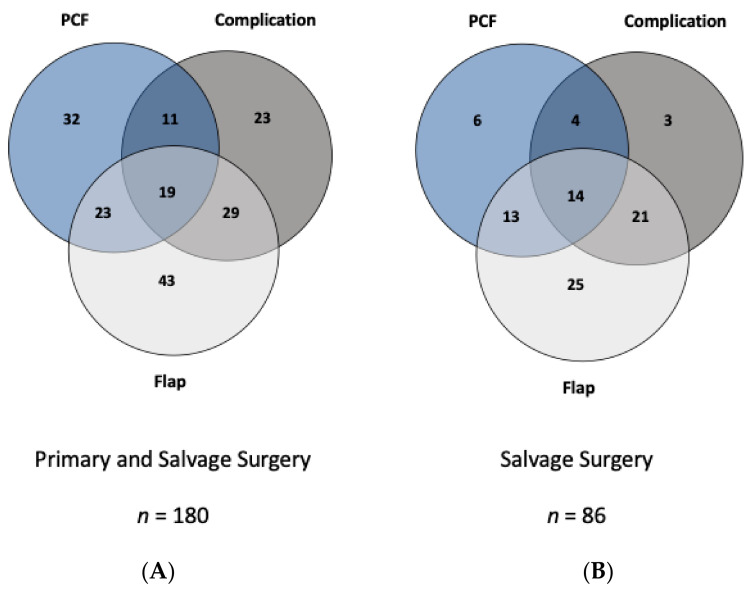
Association between complications, pharyngocutaneous fistula, and flap usage in primary and salvage surgeries. Occurrence of pharyngocutaneous fistula (PCF), short-term complications, and usage of flap during first surgery are illustrated in the Venn diagrams for the whole cohort (**A**) and those patients after salvage surgery (**B**). Using this type of illustration, the interconnections between flap use, PCF, and other complications are better intelligible. As exemplified for cohort A, 71 out of 114 patients who underwent flap reconstruction experienced postoperative complication 16.1% (*n* = 29), PCF formation 12.8% (*n* = 23), or both 10.6% (*n* = 19). Conversely, in group (**B**), 14 (16.3%) patients underwent flap reconstruction and experienced both PCF formation and postoperative complication.

**Figure 4 cancers-13-01474-f004:**
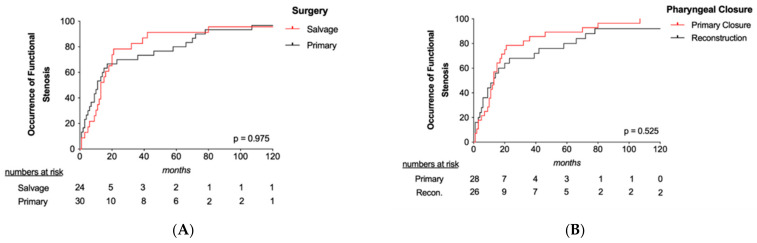
Occurrence of stenosis. Kaplan–Meier plots illustrate that the occurrence of stenosis was not influenced by surgery (**A**) or type of pharyngeal reconstruction (**B**).

**Figure 5 cancers-13-01474-f005:**
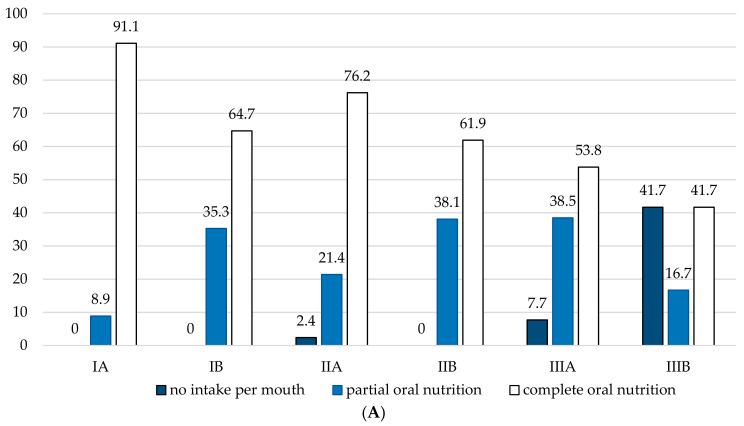
Functional outcomes according to the Vienna Laryngopharyngectomy Classification system. Charts showing functional outcomes for (**A**) oral nutrition (*p* < 0.0001) and (**B**) voice rehabilitation (*p* = 0.573) depending on the extent of ablative surgery (I: laryngectomy; II: laryngectomy with partial pharyngectomy; III: laryngectomy with total pharyngectomy) and whether each surgery was performed in salvage situation (A: non-salvage; B: salvage). Values are given in percentage.

**Table 1 cancers-13-01474-t001:** Patient cohort.

	Overall	Primary	Salvage	
Variables	*n (%)*	*n (%)*	*n (%)*	*p*-Value
**Sex**	258 (100)	161 (62.4)	97 (37.6)	
Female	28 (10.9)	16 (9.9)	12 (12.4)	
Male	230 (89.1)	145 (90.1)	85 (87.6)	0.543 ^a^
**Age** (mean ± SD)	59.2 ± 9.2	58.5 ± 9.2	60.4 ± 9.1	0.103 ^b^
**Charlson Comorbidity Index**	2.3 ± 1.8	2.2 ± 1.7	2.4 ± 1.7	0.334 ^b^
**Smoking**				
No history of smoking	49 (19)	27 (16.8)	22 (22.7)	
Active smoker	153 (59.3)	109 (67.7)	44 (45)	
Former smoker	56 (21.7)	25 (15.5)	31 (32)	**0.001** ^a^
**Primary Tumor Site**				
Larynx	138 (53.5)	86 (53.4)	52 (53.6)	
Hypopharynx	120 (46.5)	75 (46.6)	45 (46.4)	0.976 ^a^
**T-Stage ***				
T1	10 (3.9)	5 (3.1)	5 (5.2)	
T2	33 (12.8)	18 (11.2)	15 (15.5)	
T3	88 (34.1)	65 (40.4)	23 (23.7)	
T4a	117 (45.3)	74 (44.1)	46 (47.4)	0.093 ^a^
**N-Stage ***				
N0	125 (48.4)	57 (35.4)	68 (70.1)	
N1	22 (8.5)	14 (8.7)	8 (8.2)	
N2	91 (35.3)	81 (48.4)	13 (13.4)	
N3	10 (3.9)	10 (6.2)	0 (0.0)	**<0.001** ^a^
**Indication for TL**				
Primary Tumor	151 (58.5)	147 (91.3)	4 (4.1)	
Recurrence	89 (34.5)	11 (6.8)	78 (80.4)	
Residual Tumor	11 (4.3)	3 (1.8)	8 (8.2)	
Dysfunctional Larynx	7 (2.7)	0 (0.0)	7 (7.2)	**<0.001** ^a^
**Type of Surgery**				
TL	85 (32.2)	61 (37.9)	24 (24.7)	
TLPP	101 (39.1)	63 (39.1)	38 (39.2)	
TLTP	72 (27.9)	37 (23)	35 (36.1)	**0.032** ^a^

***** 3 patients underwent laryngectomy due to chondroma/chondrosarcoma and 7 due to dysfunctional larynx, which were not respected in the T- and N-Stage. ^a^ Chi-square test; ^b^ unpaired Student’s t-test.

**Table 2 cancers-13-01474-t002:** Surgical ablation, reconstruction and complications according to the Vienna Larygopharyngectomy Classification system.

	Laryngectomy		Laryngectomy with Partial Pharyngectomy		Laryngectomy with Total Pharyngectomy		
	IA	IB	IIA	IIB	IIIA	IIIB	
Variables	*n* (%)		*n* (%)		*n* (%)		*p*
**Used Flaps**							
SAFF	0 (0)	0 (0)	0 (0)	8 (22.9)	2 (4.8)	1 (2.4)	
ALT	0 (0)	0 (0)	0 (0)	1 (2.9)	2 (4.8)	4 (9.5)	
RFFF	0 (0)	0 (0)	1 (11.1)	0 (0)	0 (0)	0 (0)	
Jejunum	0 (0)	0 (0)	0 (0)	0 (0)	31 (73.8)	10 (23.8)	
Jejunum + PMMF	1 (16.7)	1 (5.9)	1 (11.1)	1 (2.9)	1 (2.4)	16 (38.1)	
RFFF+ PMMF	0 (0)	0 (0)	0	0 (0)	0 (0)	3 (7.1)	
PMMF	5 (83.3)	16 (94.1)	7 (77.8)	25 (71.4)	6 (14.3)	8 (19.1)	
**Salivary tube** *							
No	57 (93.4)	19 (79.2)	58 (92.1)	23 (60.5)	35 (94.6)	25 (71.4)	
Yes	1 (1.6)	5 (20.8)	1 (1.6)	14 (36.8)	2 (5.4)	8 (22.9)	**<0.001 ^a^**
**Complications**							
No	50 (82)	19 (79.2)	45 (71.4)	21 (55.3)	26 (70.3)	15 (42.9)	
Yes	11 (18)	5 (20.8)	18 (28.6)	17 (44.7)	11 (29.7)	20 (57.1)	**0.001 ^a^**
Bleeding	4 (36.4)	1 (20)	1 (5.6)	6 (35.3)	3 (27.3)	5 (25)	
Wound healing	1 (9.1)	1 (20)	8 (44.4)	2 (11.8)	0 (0)	2 (10)	
Infection	0 (0)	1 (20)	2 (11.1)	0 (0)	2 (18.2)	0 (0)	
Hematoma; Chyle; Seroma	4 (36.4)	2 (40)	5 (27.8)	5 (29.4)	1 (9.1)	1 (5)	
Stroke; CPR; Heart attack	2 (18.2)	0 (0)	2 (11.1)	0 (0)	0 (0)	0 (0)	
Donor site	0 (0)	0 (0)	0 (0)	1 (5.9)	2 (18.2)	6 (30)	
Flap failure	0 (0)	0 (0)	0 (0)	1 (5.9)	1 (9.1)	4 (20)	
Combined	0 (0)	0 (0)	0 (0)	2 (11.8)	2 (18.2)	2 (10)	**0.005 ^a^**
Surgical complications	9 (14.8)	4 (16.7)	14 (22.2)	17 (44.7)	9 (24.3)	21 (60)	
General complications	2 (3.3)	1 (4.2)	4 (6.3)	0 (0)	4 (10.8)	2 (5.7)	**0.001 ^a^**
**Revision/Follow-up surgeries**							
No	52 (85.2)	17 (70.8)	51 (81)	25 (65.8)	27 (73)	17 (48.6)	
Yes	9 (14.8)	7 (29.2)	12 (19)	13 (34.2)	10 (27)	18 (51.4)	**0.002 ^a^**
**PCF**							
No	47 (77)	16 (66.7)	43 (68.3)	21 (55.3)	23 (62.2)	23 (65.7)	
Yes	14 (23)	8 (33.3)	20 (31.7)	17 (44.7)	14 (37.8)	12 (34.3)	0.345 ^a^
≤30 days	11 (18)	6 (25)	18 (28.6)	16 (42.1)	7 (18.9)	9 (25.7)	0.139 ^a^
>30 days	3 (5)	2 (8.3)	2 (3.1)	1 (2.6)	7 (18.9)	3 (8.6)	**0.047 ^a^**
Persistent	0 (0)	0 (0)	2 (10)	3 (17.6)	5 (35.7)	7 (58.3)	**0.001 ^a^**

***Abbreviations:*** I, laryngectomy; II, laryngectomy with partial pharyngectomy; III, laryngectomy with total pharyngectomy ; A, non-salvage; B, salvage; SAFF, serratus anterior free flap; ALT, anterolateral thigh flap; RFFF, radial forearm free flap; PMMF. Pectoralis major muscle flap; PCF, pharyngocutaneous fistula; * missing data (*n* = 10); ^a^ Chi-square test.

**Table 3 cancers-13-01474-t003:** Pharyngocutaneous fistula.

	Univariate Analysis		
Variables	OR	*p*-Value	95% CI
Sex (Female)	2.24	**0.046**	1.01–4.94
Age (≤60 a)	0.88	0.632	0.52–1.48
Preoperative BMI (≤24.2)	1.21	0.588	0.60–2.47
Comorbidity Index (≤2.0)	0.90	0.707	0.53–1.54
Tumor site (Larynx)	0.75	0.388	0.38–1.45
T1–T2 vs. T3–T4a	1.28	0.485	0.64–2.53
N− vs. N+	0.88	0.621	0.51–1.49
Free Flap (No)	0.64	0.110	0.37–1.10
Pharynx reconstruction (No)	0.73	0.261	0.42–1.23
No reconstruction with jejunum	2.11	0.105	0.85–5.18
Pharyngeal closure (unilayer)	1.13	0.699	0.62–2.05
Pharyngeal closure (T-form)	1.12	0.720	0.61–2.03
Salivary tube (No)	0.54	0.116	0.25–1.16
Non-Salvage Surgery	0.69	0.169	0.41–1.17
PORT (No)	1.44	0.174	0.85–2.42
Class. (continuous)	1.14	0.094	0.98–1.33
Complications (No)	0.79	0.396	0.45–1.37

Univariate binary logistic regression analysis was performed to evaluate whether or not clinical variables represent significant predictors for developing a salivary fistula after laryngectomy. Median was used for metric variables (age, preoperative BMI, Comorbidity Index) to dichotomize patients into subgroups. OR, odds ratio; 95% CI, 95% confidence interval; Class. (continuous), groups IA/IB,IIA/IIB,IIIA/IIIB according to the Vienna Laryngopharyngectomy Classification system.

**Table 4 cancers-13-01474-t004:** Complete oral nutrition, G-tube dependence, and pharyngeal stenosis.

	Univariate Analysis			Multivariate Analysis		
Complete Oral Nutrition	OR	*p*-Value	95% CI	OR	*p*-Value	95% CI
Sex (Female)	0.60	0.322	0.22–1.65			
Age (≤60 a)	0.74	0.389	0.37–1.47			
Preoperative BMI (≤24.2)	0.58	0.140	0.28–1.19			
Comorbidity Index (≤2.0)	0.62	0.202	0.30–1.29			
Tumor site (Larynx)	1.84	0.082	0.89–3.51			
T1-T2 vs. T3-T4a	1.36	0.547	0.50–3.64			
N- vs. N+	1.82	0.099	0.89–3.72			
Pharynx reconstruction (No)	4.72	**<0.001**	2.25–9.91	4.44	**0.042**	1.05–18.71
Salivary tube (No)	1.53	0.383	0.59–3.98			
Salvage surgery (No)	2.42	**0.015**	1.19–4.94	1.97	0.134	0.81–4.76
PORT (No)	0.65	0.224	0.33–1.30			
Class. (continuous)	0.63	**<0.001**	0.50–0.79	0.99	0.945	0.64–1.53
PCF (No)	3.13	**0.002**	1.54–6.37	3.29	**0.003**	1.49–7.28
Short-term complications (No)	3.56	**0.001**	1.68–7.55	3.47	**0.004**	1.48–8.14
**G-Tube Dependence**						
Sex (Female)	4.31	**0.013**	1.36–13.66	2.06	0.291	0.54–7.88
Age (≤60 a)	1.53	0.361	0.62–3.81			
Preoperative BMI (≤ 24.2)	2.48	0.085	0.88–6.96			
Comorbidity Index (≤ 2.0)	1.55	0.375	0.59–4.04			
Tumor site (Larynx)	0.27	**0.006**	0.11–0.69	0.68	0.541	0.20–2.33
T1-T2 vs. T3-T4a	1.96	0.216	0.68–5.68			
N- vs. N+	0.60	0.280	0.24–1.51			
Pharyngeal reconstruction (No)	0.21	**0.001**	0.08–0.53	0.84	0.849	0.15–4.90
Salivary tube (No)	0.51	0.245	0.16–1.59			
Salvage surgery (No)	0.42	0.061	0.17–1.04			
PORT (No)	1.56	0.330	0.64–3.84			
Class. (continuous)	1.77	**<0.001**	1.30–2.40	1.48	0.222	0.79–2.78
PCF (No)	0.20	**0.001**	0.08–0.51	0.21	**0.004**	0.07–0.60
Short-term complications (No)	0.30	**0.011**	0.12–0.76	0.37	0.075	0.12–1.10
**Pharyngeal Stenosis**						
Sex (Female)	4.81	**0.003**	1.69–13.72	3.56	0.074	0.88–15.02
Age (≤60 a)	1.18	0.660	0.57–2.44			
Preoperative BMI (≤ 24.2)	2.97	**0.014**	1.24–7.11	1.44	0.482	0.52–4.07
Comorbidity Index (≤ 2.0)	1.18	0.665	0.56–2.52			
Tumor site (Larynx)	0.26	**0.001**	0.12–0.56	0.84	0.753	0.28–2.54
T1-T2 vs. T3-T4a	1.98	0.140	0.80–4.88			
N- vs. N+	0.45	**0.036**	0.21–0.95	0.66	0.396	0.26–1.71
Pharyngeal reconstruction (No)	0.18	**<0.001**	0.08–0.40	0.90	0.900	0.19–4.40
Salivary tube (No)	1.11	0.847	0.37–3.23			
Salvage surgery (No)	0.58	0.171	0.27–1.26			
PORT (No)	0.88	0.741	0.41–1.88			
Class. (continuous)	1.82	**<0.001**	1.40–2.37	1.57	0.085	0.94–2.64
PCF (No)	0.77	0.490	0.36–1.63			
Short-term complications (No)	0.45	0.051	0.20–1.00			

Uni- and multivariate binary logistic regression analyses were performed to evaluate whether clinical variables represent significant predictors for successful oral nutrition, temporary gastrostomy tube dependence, and occurrence of pharyngeal stenosis. The median was used for metric variables (age, preoperative BMI, Comorbidity Index) to dichotomize patients into subgroups. OR, odds ratio; 95% CI, 95% confidence interval; Class. (continuous), groups IA/IB,IIA/IIB,IIIA/IIIB according to Vienna Laryngopharyngectomy Classification system; PORT, postoperative radiotherapy; PCF, pharyngocutaneous fistula.

## Data Availability

Data is contained within the article and is available on request.

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
