# Peer review of "A New Classification System to Predict Functional Outcome after Laryngectomy and Laryngopharyngectomy"

_cancers, 2021, doi:10.3390/cancers13061474_

Round 1

Reviewer 1 Report

This is a well written paper on complications and functional outcome after primary vs salvage surgery for laryngeal/hypopharyngeal cancer. Overall the paper is well written, provides useful information and certainly deserves publication. 

Abstract: the abstract does not fully reflect the title of the paper as f.i. no data on voice outcome are reported.  More information on functional outcome rather than complications should be reported in the abstract

Introduction: well written; I would reccomned to include among the references the European white paper on dysphagia in Head and Neck cancer pts (Baijens et al, Eur Arch Otorhinolaryngol 2021)

Methods: the authors did not use validated scales for oral nutrition such as the FOIS (Functional Outcome for Oral Intake Scale): this should be reported as a limitation

Results: in fig. 3 I would add % in brackets in order to facilitate comparison between fig a and b

Discussion: page 16, line 406: I would soften the statement that g-tube was rarely required  

Reviewer 2 Report

I give my congratulations to the authors for their effort in writing such a well-structured and interesting article.

I would suggest checking and correct minor English grammar errors (i.e. line 73 Population study instead of Study population; line 234 with complications in 28.6% and 29.7% of the cases (put "the"); line 235 higher risk of complications instead of using "for"; ...)

Could you add adjuvant treatment data, please? I think this would be worthy of investigation in terms of post-operative and late complication rates onset. 

Thank you

Reviewer 3 Report

Thx for allowing me to review this manuscript. This is a retrospective study using instititutional data to be able to predict compications and long term function in patients undergoing laryngectomy with partial or total pharyngectomy. This is an important topic as the changes to patients lives after surgery affect patients ability to swallow and verbally communicate and can thus be used to guide patients in decision making and managing expectations.

The introduction is well written and covers the most important topics.The discussion section is succinct and also easy to follow. The result section is somewhat extensive and can be shortened given that most of the data is presented in tables.

Although I think in general the paper is worthy of publication I believe the following can strengthen the paper:

The definition of partial pharyngectomy is poorly (if at all defined) This needs better clarity especially since some partial pharyngectomies were closed primarily. Was this based on how much pharynx was resected or how much was remaining? By default this becomes very subjective unless measurements were made intraoperatively. A more stringent  definitin may be if patient required an inlay epithialized flap as definition of partial pharyngectomy.

The authors have classified the function based on salvage or non salvage. However, I do not see how many of primary laryngectomy patients recieved radiation. This difference may be completely driven by affect of radiation, especially since many patients develop functional issues after a year. It may be worthwhile comparing the sx+(c)rt vs (c)rt+sx as a subset. I would expect the best function in single modality surgical patients without need for reconstruction.

The authors have included patients over a very long time span. The introduction of IMRT during this time frame may have changed treatment. Did the authors explore whether complications or functional outcomes changed over time? Equally so for the use of speech valves?

The reconstructive practice with a large majority of reconstructed patients having a jejunal flap is likely poorly representative of practice in most other institutions. I'd mention this more explicitly in the limitation section as to the generalizability of these findings.
